

**Particle acidity and sulfate production during severe haze events in China**
**cannot be reliably inferred by assuming a mixture of inorganic salts**
Gehui Wang[1,2,3], Fang Zhang[4,5], Jianfei Peng[5,6], Lian Duan[5,7], Yuemeng Ji[5,8], Wilmarie
Marrero-Ortiz[5], Jiayuan Wang[2], Jianjun Li[2], Can Wu[2], Cong Cao[2], Yuan Wang[9], Jun Zheng[10],
Jeremiah Secrest[5], Yixin Li[5], Yuying Wang[4,5], Hong Li[11], Na Li[5,12], and Renyi Zhang[5,6*]
[1] Key Laboratory of Geographic Information Science of the Ministry of Education, School of
Geographic Sciences, East China Normal University, Shanghai 200241, China
[2] State Key Laboratory of Loess and Quaternary Geology, Institute of Earth Environment, China
Academy of Sciences, Xi'an 710061, China
[3] Center for Excellence in Regional Atmospheric Environment, Institute of Urban Environment,
Chinese Academy of Science, Xiamen, China
[4] Beijing Normal University, Beijing 100875, China
[5] Departments of Atmospheric Sciences and Chemistry, Texas A&M University, College Station,
TX, 77843, USA
[6] State Key Joint Laboratory of Environmental Simulation and Pollution Control, College of
Environmental Sciences and Engineering, Peking University, Beijing 100871, China
[7] East China University of Science and Technology, Shanghai, China
[8] School of Environmental Science and Engineering, Institute of Environmental Health and
Pollution, Control, Guangdong University of Technology, Guangzhou 510006, China
[9] Jet Propulsion Laboratory, California Institute of Technology, Pasadena, CA 91125, USA
[10] Jiangsu Key Laboratory of Atmospheric Environment Monitoring and Pollution Control,
Nanjing University of Information Science & Technology, Nanjing 210044, China
[11] State Key Laboratory of Environmental Criteria and Risk Assessment, Chinese Research
Academy of Environmental Sciences, Beijing 100012, China
[12] Key Laboratory of Songliao Aquatic Environment, Jilin Jianzhu University, Changchun,
130118, China
*Corresponding author:
Prof. Renyi Zhang, E-mail: renyi-zhang@tamu.edu



**Abstract**: Atmospheric measurements showed rapid sulfate formation during severe haze
episodes in China, with fine particulate matter (PM) consisting of a multi-component mixture
that is dominated by organic species. Several recent studies using the thermodynamic model
estimated the particle acidity and sulfate production rate, by treating the PM exclusively as a
mixture of inorganic salts dominated by ammonium sulfate and neglecting the effects of organic
compounds. Noticeably, the estimated pH and sulfate formation rate during pollution periods in
China were highly conflicting among the previous studies. Here we show that a particle mixture
of inorganic salts adopted by the previous studies does not represent a suitable model system and
that the acidity and sulfate formation cannot be reliably inferred without accounting for the
effects of multi-aerosol compositions during severe haze events in China. Our laboratory
experiments show that $SO_2$ oxidation by $NO_2$ with $NH_3$ neutralization on fine aerosols is
dependent on the particle hygroscopicity, phase-state, and acidity. Ammonium sulfate and oxalic
acid seed particles exposed to vapors of $SO_2$, $NO_2$, and $NH_3$ at high relative humidity (RH)
exhibit distinct size growth and sulfate formation. Aqueous ammonium sulfate particles exhibit
little sulfate production because of high acidity, in contrast to aqueous oxalic acid particles with
significant sulfate production because of low acidity. Our field measurements demonstrate
significant contribution of water-soluble organic matter to fine PM in China and indicate that the
use of oxalic acid in laboratory experiments is representative of ambient organic dominant
aerosols. While the particle acidity cannot be accurately determined from field measurements or
calculated using the thermodynamic model, our results reveal that the pH value of ambient
organics-dominated aerosols is sufficiently high to promote efficient $SO_2$ oxidation by $NO_2$ with
$NH_3$ neutralization under polluted conditions in China.




## 1. Introduction


Atmospheric measurements have demonstrated rapid sulfate production during severe haze
events in China (Guo et al., 2014; Wang et al., 2014; Zhang et al., 2015; Cheng et al., 2016;
Wang et al., 2016). For example, Wang et al. (2016) showed that during pollution episodes in
Xi'an of China the $SO_4^{2-}$ mass concentration increased markedly from less than 10, 10-20, to
greater than 20 μg m$^{-3}$, with the corresponding increases in the mean $PM_{2.5}$ mass concentrations
from 43, 139, to 250 μg m$^{-3}$ from clean, transition, to polluted periods, respectively. Among the
$PM_{2.5}$ species in Xi'an, organic matter (OM), nitrate ($NO_3^-$), and $SO_4^{2-}$ were most abundant, with
the mass fractions of 55%, 14%, and 14%, respectively, during the polluted period. In addition,
the work by Wang et al. (2016) demonstrated that the molar ratio of $SO_4^{2-}$ to $SO_2$, which reflects
sulfur partitioning between the particle and gas phases, exhibited an exponential increase with
relative humidity (RH), with the values of less than 0.1 at RH < 20% to 1.1 at RH > 90% in
Xi'an. Similar evolutions in $SO_4^{2-}$ mass concentrations and the molar ratio of $SO_4^{2-}$ to $SO_2$ were
shown during the pollution development in Beijing (Sun et al., 2013; Wang et al., 2014; Wang et
al., 2016). The rapid sulfate formation measured in China could not be explained by current
atmospheric models and suggested missing sulfur oxidation mechanisms (Wang et al., 2014).
Typically, high sulfate levels during haze events in China occurred concurrently with elevated
RH, $NO_x$, and $NH_3$ (Wang et al., 2014; Zhang et al., 2015; Wang et al., 2016), implicating an
aqueous sulfur oxidation pathway. On the basis of complementary field and experimental
measurements, Wang et al. (2016) concluded that the aqueous oxidation of $SO_2$ by $NO_2$ is key to
efficient sulfate formation, but is only feasible under two atmospheric conditions, i.e., on fine
aerosols with high RH and $NH_3$ neutralization or under cloud conditions.
Several recent studies estimated the particle acidity and aqueous sulfate production during



severe haze events in China using the thermodynamic model (Cheng et al., 2016; Guo et al.,
2017; Liu et al., 2017). For example, Cheng et al. (2016) estimated a pH range of 5.4 to 6.2 using
a thermodynamic model (ISORROPIA-II) in Beijing. On the basis of their estimated pH and the
previous experimental rates of $SO_2$ oxidation by $NO_2$ and the Henry's Law constants for sulfur
dioxide ($SO_2$), bisulfite ($HSO_3^-$), and sulfite ($SO_3^{2-}$) from the literature (Lee and Schwartz, 1983;
Clifton et al., 1988; Seinfeld and Pandis, 2006), the authors derived a sulfate production rate and
concluded that reactive nitrogen chemistry in aerosol water explained the sulfate formation
during polluted periods in Beijing. In contrast, other recent studies by Guo et al. (2017) and Liu
et al. (2017) adopted the similar method as Cheng et al. (2016), but reported significantly
different values of pH and the sulfate formation rates by the aqueous $SO_2$ oxidation by $NO_2$ in
China. Those two later studies determined a pH range of 3.0-4.9 and suggested that fine particles
were moderately acidic and the aqueous $SO_2$ oxidation by $NO_2$ was unimportant during severe
wintertime haze periods in China.

In this article, we conducted laboratory measurements of the hygroscopicity for oxalic acid

particles and particle growth of ammonium sulfate particles upon exposure to $SO_2$, $NO_2$, and
$NH_3$ at high RH conditions, in order to evaluate the dominant factors regulating the aqueous
oxidation of $SO_2$ by $NO_2$. In addition, field measurements of chemical compositions of water-
soluble fraction for fine PM (including oxalic acid) in Beijing, Hebei Province, and Xi'an of
China were performed during the winter haze episodes, showing significantly enriched water-
soluble organic matter (WSOM). The implications for the multi-aerosol chemical compositions
on the pH value and sulfate production during winter pollution periods in China are discussed.
**2.    Methods**
**2.1    Aqueous phase oxidation of SO₂ by NO₂ in an environmental chamber**




The experimental method using the environmental chamber has been discussed elsewhere
(Wang et al., 2016), and here we only provide a brief description. The aqueous $SO_2$ oxidation
experiments was conducted by exposing size-selected $(NH_4)_2SO_4$ seed particles to different
levels of $SO_2$, $NO_2$, and $NH_3$ at variable RH conditions in a 1 $m^3$ Teflon reaction chamber
covered with aluminum foil. A differential mobility analyzer (DMA) equipped with a
condensation particle counter (CPC) was used to measure the particle growth in diameter, in
order to determine sulfate formation on seeded particles (Wang et al., 2016).
**2.2    Measurement of hygroscopic growth factor of oxalic acid**
Hygroscopic growth factor (HGF) of oxalic acid was measured according to the method
previously discussed (Khalizov et al., 2009; Pagels et al., 2009). Briefly, a hygroscopicity
tandem differential mobility analyzer (HTDMA) coupled to a condensation particle counter
(CPC, TSI 3762) was used for the HGF measurement. Size-selected oxalic acid particles with the
dry diameter of 100 nm were exposed to increasing RH from 8% to 92% with a step range from
1%-10%. HGF is defined as the ratio of oxalic acid particle diameter ($D_p$) measured by the
second DMA at an elevated RH to the initial diameter ($D_0 = 100$ nm) of the particles selected by
the first DMA at the dry conditions of RH = 8% (Peng et al., 2016).
**2.3    Chemical composition of $PM_{2.5}$ in Beijing, Hebei Province, and Xi'an, China**
$PM_{2.5}$ samples were collected onto pre-baked ($450^o$C for 6 hr) quartz fiber filter by using a
high-volume air sampler with an airflow rate of 1.03 $m^3$ $min^{-1}$. The sample collection in Xi'an
was performed on the roof of a three-story building in the urban center with a 1-hour interval for
each sample during the winter of 2012 (Wang et al., 2016). The sample collection in Beijing was
conducted during the winter of 2016 on the roof of a four-story building on the campus of China
Research Academy of Environmental Sciences, which is located at the northern part of Beijing.





The $PM_{2.5}$ samples in Hebei Province were collected during the winter of 2016 on the roof of a
three-story building on the campus of the Institute of Hydrology and Environmental Geology,
which is located in Zhengding County of Hebei Province. Both sample collections in Beijing and
Hebei Province were performed on a day/night basis. After collection, all samples were sealed
individually in an aluminum foil bag and stored in a freezer below -18$^\circ$C prior to analysis.

The detailed procedures for the analysis of inorganic ions and water-soluble organic matter

(WSOM) in aerosols have been reported elsewhere (Wang et al., 2009; Wang et al., 2010; Wang
et al., 2017). Briefly, one part of the filter sample (area about 5 cm$^2$) was divided into several
pieces, extracted with Mili-Q pure water, and determined for WSOM and inorganic ions by
using Shimadzu TOC-L CPH analyzer and Dionex-600 ion chromatography, respectively.
Oxalic acid in $PM_{2.5}$ was analyzed according to Wang et al. (2002) and Cheng et al. (2015). One
part of the filter sample was extracted with Milli-Q water, concentrated to dryness, and reacted
with 14% BF3/butanol at 100$^\circ$C for 1 hr. After the reaction, the derivatized sample was extracted
with hexane for three times and concentrated into 1 mL. Oxalic acid in the samples was
identified by gas chromatography–mass spectrometry (GC–MS) and quantified by gas
chromatography (Agilent GC7890A).
**3.    Results**
**3.1    Aqueous oxidation of $SO_2$ by $NO_2$ with $NH_3$ neutralization**

We first evaluated the factors controlling the aqueous phase oxidation of $SO_2$ by $NO_2$ using

the environmental chamber method. The evolution in the size of ammonium sulfate particles
after exposure to $SO_2$, $NO_2$, and $NH_3$ at different RH and $SO_2$ levels is shown in Figure 1. In our
experiments, monodisperse particles with the initial dry particle size ranging from 50 to 70 nm
were selected for the exposure, and two different $SO_2$ concentrations (37.5 and 375 parts per



billions or ppb) were used. RH was maintained at a level of 80-98%, above the deliquescence
point (79%) of ammonium sulfate (Qiu and Zhang, 2013) to ensure aqueous particles. As is
shown in Figure 1, the size of $(NH_4)_2SO_4$ particles remains nearly invariant (within the
experimental uncertainty) after exposure to $SO_2$, $NO_2$, and $NH_3$. A 10-fold increase in the $SO_2$
concentration has little effect on the growth of $(NH_4)_2SO_4$ particles. These results illustrate that
sulfate production is insignificant and $SO_2$ cannot be efficiently oxidized by $NO_2$ in the presence
of $NH_3$ on aqueous ammonium sulfate particles. The measurement of negligible growth for
$(NH_4)_2SO_4$ particles exposed to $SO_2$, $NO_2$, and $NH_3$ at high RH is in contrast to the previous
work by Wang et al. (2016), which showed large size growth and significant sulfate production
for oxalic acid particles with $NH_3$ neutralization and under high RH conditions.

To gain an insight into such a difference in the size growth between $(NH_4)_2SO_4$ and oxalic

acid particles, we measured the hygroscopic growth of oxalic acid particles. Figure 2 displays the
measured hygroscopic growth factor (HGF) of oxalic acid, showing an exponential increase with
an increase in RH. The measured HGF value is close to unity at RH < 40% and increases from
1.1 at RH = 60% to 1.5 at RH = 90%. Our measured HGF for oxalic acid is consistent with the
previous studies by Prenni et al. (2001) and Mikhailov et al. (2009). On the other hand, another
earlier experimental study showed little growth for oxalic acid particles under high RH
conditions (Peng et al., 2001). The measurements of HGF also provide information on the
particle phase-state. As evident from Figure 2, oxalic acid particles mainly exist in a non-
aqueous phase at RH < 40% but in the aqueous phase at RH > 60%.

Our present experiments of aqueous oxidation of $SO_2$ by $NO_2$ were performed at similar

conditions as those by Wang et al. (2016), i.e., with comparable concentrations for $SO_2$, $NO_2$,
and $NH_3$ and in the same phase-state (aqueous) for the particles. On the other hand, the particle





acidity is clearly distinct between the two studies. Our present experiment is characterized by a
lower pH value, since ammonium sulfate is rather acidic. For example, the pH value of 0.1M
$(NH_4)_2SO_4$ solution is 5.5. The overall aqueous reaction between $SO_2$ and $NO_2$ in the presence of
$NH_3$ is suggested as the following (Wang et al., 2016),

$2NH_3(g) + SO_2(g) + 2NO_2(g) + 2H_2O(aq) \rightarrow 2NH_4^+(aq) + SO_4^{2-}(aq) + 2HONO(g)$        (1)

Since the solubility of $SO_2$ and $NO_2$ decreases markedly with increasing particle acidity
(Seinfeld and Pandis, 2006; Zhang et al., 2015), the heterogeneous reaction between $SO_2$ and
$NO_2$ is prohibited on acidic $(NH_4)_2SO_4$ particles. On the other hand, under the experimental
conditions by Wang et al. (2016), the heterogeneous reaction between oxalic acid and $NH_3$
occurred on aqueous particles in the presence of $NH_3$, yielding ammonium oxalate. The
ammonium oxalate is less acidic than ammonium sulfate. The pH value of 0.1 M ammonium
oxalate is 6.5, which is one unit higher than that of ammonium sulfate. As a result, $SO_2$ readily
dissolves into aqueous ammonium oxalate particles and is oxidized by $NO_2$ into $SO_4^{2-}$, which is
consequently neutralized by $NH_3$ to produce $(NH_4)_2SO_4$. The resulting aqueous ammonium
oxalate/$(NH_4)_2SO_4$ particles exhibit a lower acidity than that of $(NH_4)_2SO_4$ particles, responsible
for a significant growth in the dry particle size and sulfate formation for the previous
experiments by Wang et al. (2016).

Hence, the experimental studies of our present work and that by Wang et al. (2016) reveal

that sulfate production on fine particles is dependent on several factors, including the particle
hygroscopicity, phase-state, acidity, and RH, in addition to the gaseous concentrations of $SO_2$,
$NO_2$, and $NH_3$. These experimental results indicate that the acidity and sulfate formation are
distinct for organic seed and ammonium sulfate seed particles. While oxidation of $SO_2$ by $NO_2$
on aqueous $(NH_4)_2SO_4$ particles does not represent a viable mechanism because of a higher





acidity, significant sulfate production occurs on oxalic acid particles because of a lower acidity.
**3.2    Field measurements of WSOM in China**

Atmospheric measurements have shown that the occurrence of severe haze episodes in

China is accompanied with high RH conditions and $PM_{2.5}$ particles consist of large amounts of
secondary organic and inorganic compounds. We present additional field measurements of the
chemical composition of $PM_{2.5}$ in Beijing, Hebei Province, and Xi'an of China. Figure 3 shows
that the wintertime $PM_{2.5}$ samples collected at the three locations. It is evident that WSOM is
considerably enriched and their concentrations are comparable to those of the total inorganic ions
(Figure 3a and b). For example, the mass concentration of WSOM ranges from 10 to 60 µg m$^{-3}$
in Beijing and Hebei Province during the winter of 2016 and from 10 to 180 µg m$^{-3}$ in Xi'an
during the winter of 2012 (Figure 3c and d, respectively). In addition, the variation of WSOM
displays a temporal pattern similar to that of oxalic acid, with a linear correlation coefficient of
0.79, 0.88 and 0.72 in Beijing, Hebei Province, and Xi'an, respectively (Figure 3e and f). The
mass concentration of oxalic acid in fine PM during the haze episodes is about 500 ng m$^{-3}$ in
Beijing and Hebei Province (Figure 3e) and more than 2000 ng m$^{-3}$ in Xi'an (Figure 3f). Hence,
our field measurements indicate that oxalic acid represents one of the most abundant WSOM in
the aerosol-phase. Oxalic acid, a secondary product formed from the photochemical oxidation of
volatile organic compounds, has been also shown to exist with large abundance in China (Wang
et al., 2012; Cheng et al., 2013; Meng et al., 2014; Kawamura and Bikkina, 2016). In addition,
the previous field measurements also revealed that WSOM in China is not only enriched in
carboxylic acids (including oxalic acid) but also in other organic species, including carbonyls,
amines, and water-soluble nitrogen-containing organic compounds (Wang et al., 2010, 2013;
Zheng et al., 2015; Yao et al., 2016; Liu et al., 2017). The dominant organic acids and bases



indicate that haze particles in China are multi-component in nature and the estimations of the
particle acidity (or pH) and the sulfate production rate need to take into account of the effects of
organic species, in addition to inorganic ions.
**4.        Discussions**

Several recent studies using the thermodynamic models (Wexler and Clegg, 2002;

Fountoukis and Nenes, 2007) estimated the particle acidity and sulfate production during
pollution episodes in China (Cheng et al., 2016; Guo et al., 2017; Liu et al., 2017). Those
previous studies treated the PM exclusively as a mixture of inorganic salts dominated by
ammonium sulfate and neglected the effects due to the presence of organic compounds.
Apparently, the conclusions by those modeling studies hinge on the validity of several critical
assumptions in their analyses, including the application of the thermodynamic model, the
accuracy in determining the aerosol water content (AWC), and the applicability of the earlier
experimental measurements for the aqueous oxidation of $SO_2$ by $NO_2$ to atmospheric conditions.

Estimation of the pH values using the thermodynamic models is typically of considerable

uncertainty, because of several intricate difficulties. For example, the ISORRPIA-II model
includes two modes, i.e., metastable (aerosols are assumed to be in the liquid-phase only and
may reach supersaturation) and stable (aerosols are assumed in the liquid- and solid phases that
are in equilibrium) (Guo et al., 2017). Since the thermodynamic model is established on the basis
of the equilibrium principles, its application to non-equilibrium conditions needs to be rigorously
assessed. Also, the phase (e.g., liquid, amorphorous, or crystalline) and mixing state of ambient
aerosols are highly complex because of the presence of multi-component organic and inorganic
species (Qiu and Zhang, 2013; Zhang et al., 2015), inevitably rendering high uncertainty in the
thermodynamic calculations.



Guo et al. (2017) suggested that the pH predictions using the metastable mode would be
more reliable than that using the stable mode, on the basis of model evaluation from measured
and predicted $NO_3^-$ and $NH_4^+$ during the winter of 2012 in Xi'an. Figure 4 compares the
concentrations of $NH_3$ (g) and aerosol species predicted by ISORROPIA-II with the field
measurements under the metastable and stable modes in Xi'an during the winter of 2012. As
evident in Figure 4a and b, $NH_3$ predicted is similar to the measured value with the metastable or
stable mode. Furthermore, the predicted concentrations of $NO_3^-$ and $NH_4^+$ using both the
metastable and stable modes are nearly identical (Figure 4c-f). Guo et al. (2017) only compared
the liquid $NH_4^+$ and $NO_3^-$ predicted by the model with the field measured aerosols composed of
both liquid and solid compounds, and their predicted concentrations were lower than those of the
measurements (see Figure S1 in Guo et al, 2017). As a result, their statement that pH prediction
with the metastable mode would be more reliable than that with the stable mode was unjustified.
Noticeably, the pH values estimated by the ISORROPIA-II model under the two modes are
significantly different, with the values of 4.57±0.40 under the metastable mode and 6.96 ±1.33
under the stable mode. Most recently, it was suggested that the large discrepancy in predicting
pH is attributable to the differences in the model assumptions (Song et al., 2018).
In addition, the pH estimation by the thermodynamic model is highly dependent on the ratio
of the concentration of hydrogen ions in the liquid-phase to AWC. Guo et al. (2017) and Liu et al.
(2017) assumed negligible particle water associated with the organic aerosol mass. Such an
assumption is clearly invalid since aerosols typically contain a large portion of WSOM in China
(Fig. 3), including organic nitrogen species (Wang et al., 2010, 2013) and acids (Wang et al.,
2006, 2009, 2010). Also, organic acids engage in particle-phase reactions with the basic species
(i.e., $NH_3$ and amines), significantly enhancing the particle hygroscopicity and reducing the



acidity (Gomez-Hernandez et al., 2016). In addition, because of their strong basicity and high

abundance, amines likely play a key role in reducing the particle-acidity in China (Wang et al.,

2010a, b; Qiu et al., 2011; Qiu and Zhang, 2012; Dong et al., 2013; Zheng et al., 2015; Yao et al.,

2016; Liu et al., 2017). Consequently, the acidity for organics-dominated aerosols is

considerably different from that of ammonium sulfate aerosols, as demonstrated in our

experimental results. While effort has been made to account for the effects of organic species on

the aerosol properties (Clegg et al., 2013), the available thermodynamic models are still

inadequate in representing complex multi-component aerosols. An inconsistency of the

ammonium–sulfate ratios using the thermodynamic models was identified in the eastern US, also

suggesting a possible role for organic species (Silvern et al., 2017).

Furthermore, the chemical mechanism leading to the aqueous conversion of $SO_2$ to sulfate

by $NO_2$ is not well understood. The previous modeling studies adopted the aqueous reaction rate

constants previously measured (Lee and Schwartz, 1983; Clifton et al., 1988), while the

applicability of the earlier experimental studies to atmospheric conditions is uncertain. For

example, Lee and Schwartz (1983) examined the oxidation of S(IV) by $NO_2$ in the liquid phase

by flowing gaseous $NO_2$ through a $NaHSO_3$ solution at a constant pH by regulating NaOH and

determined the rate constant of $1.4 \times 10^5\ M^{-1}\ s^{-1}$ at pH = 5 and with a lower limit of $2 \times 10^6\ M^{-1}\ s^{-1}$

at pH = 5.8 and 6.4 from measuring the electrical conductivity of the solution. Clifton et al.

(1988) measured the rate constant for the reaction of $NO_2$ with S(IV) over the pH range of 5.3-13,

by producing $NO_2$ from irradiation of $NaNO_2$ and $N_2O$ solutions and mixing with $Na_2SO_3$

solutions, and obtained the second-order rate constant of $1.24 \times 10^7$ and $2.95 \times 10^7\ M^{-1}\ s^{-1}$ from

the decay of $NO_2$ monitored by absorption spectroscopy. The results of the measured rate

constants between the two earlier experimental measurements differed by 1-2 orders of





magnitude (Lee and Schwartz, 1983; Clifton et al., 1988). Also, both kinetic experiments
employed bulk solutions and did not account for the gaseous uptake process (Lee and Schwartz,
1983; Clifton et al., 1988).

Wang et al. (2016) obtained the $SO_2$ uptake coefficient for sulfate production from

combined field measurements and laboratory experiments, and their laboratory experiments
using aqueous oxalic acid particles reproduced the rapid sulfate production measured under
polluted ambient conditions. The results of the $SO_2$ uptake coefficients determined by Wang et al.
(2016) are consistent with the modeling studies in quantification of the sulfate formation using
atmospheric models in China (e.g., Wang et al., 2014). On the other hand, Liu et al. (2017)
invoked the experimental work by Hung et al. (2015) as a plausible cause for rapid $SO_2$
oxidation by $O_2$ in the absence of photochemistry, but without noting the high acidity as a
necessary condition in that experimental work (i.e., pH $\leqslant$ 3). Most recently, Li et al. (2018)
suggested an indirect mechanism of $SO_2$ oxidation by $NO_2$ via $HONO/NO_2^-$ produced in fast-
hydrolytic disproportionation of $NO_2$ on the surface of $NaHSO_3$ aqueous microjets. In addition,
another recent theoretical work by Zhang et al. (2018) indicated that under weakly acidic and
neutral conditions (pH $\leq$ 7) the oxidation of $HOSO_2^-$ by dissolved $NO_2$ is a self-sustaining
process, where the produced $cis$-HONO, $HSO_4^-$ and $H_2SO_4$ promote the tautomerization from
$HSO_3^-$ to $HOSO_2^-$ as the catalysts.
**5.  Conclusions**

In this paper we have presented experimental measurements of the growth of ammonium

sulfate seed particles exposed to vapors of $SO_2$, $NO_2$, and $NH_3$ at variable RH, the HGF of oxalic
acid particles, and field measurements of WSOM for $PM_{2.5}$ during the severe haze events in
Beijing, Hebei Province, and Xi'an of China. Our experimental results reveal that sulfate



production on fine particles is dependent on the particle hygroscopicity, phase-state, and acidity,
as well as RH. The acidity and sulfate formation for ammonium sulfate seed particles are distinct
from those of oxalic acid seed particles. Aqueous ammonium sulfate particles show negligible
growth because of low pH, in contrast to aqueous oxalic acid particles with significant dry-size
increase and sulfate formation because of high pH. In addition, our atmospheric measurements
show significant concentrations of WSOM (including oxalic acid) in fine PM, indicating multi-
component haze particles in China. Our results reveal that a particle mixture of inorganic salts
adopted by the previous studies using the thermodynamic model does not represent a suitable
model system and that the particle acidity and aqueous sulfate formation rate cannot be reliably
inferred without accounting for the effects of multi-chemical compositions during severe haze
events in China. Our combined experimental and field measurements corroborate the earlier
finding that sulfate production via the particle-phase reaction involving $SO_2$ and $NO_2$ with $NH_3$
neutralization occurs efficiently on organics-dominated aerosols (Wang et al., 2016) but are in
contradiction to the most recent studies using the thermodynamic model (Guo et al., 2017; Liu et
al., 2017).

In conclusion, while the particle acidity or pH cannot be accurately determined from

atmospheric field measurements or calculated using the thermodynamic models, our combined
experimental and field results provide the compelling evidence that the pH value of ambient
organics-dominated particles is sufficiently high to promote $SO_2$ oxidation by $NO_2$ with $NH_3$
neutralization under polluted conditions in China.
**Acknowledgements**

Financial support for this work was provided by National Key R&D Plan (Quantitative

Relationship and Regulation Principle between Regional Oxidation Capacity of Atmospheric and





Air Quality (No. 2017YFC0210000), the China National Natural Science Funds for
Distinguished Young Scholars (No.41325014), the program from National Nature Science
Foundation of China (No. 41773117). This work was also supported by the Robert A. Welch
Foundation (Grant A-1417) W.M.-O. was supported by the National Science Foundation
Graduate Research Fellowship Program.



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




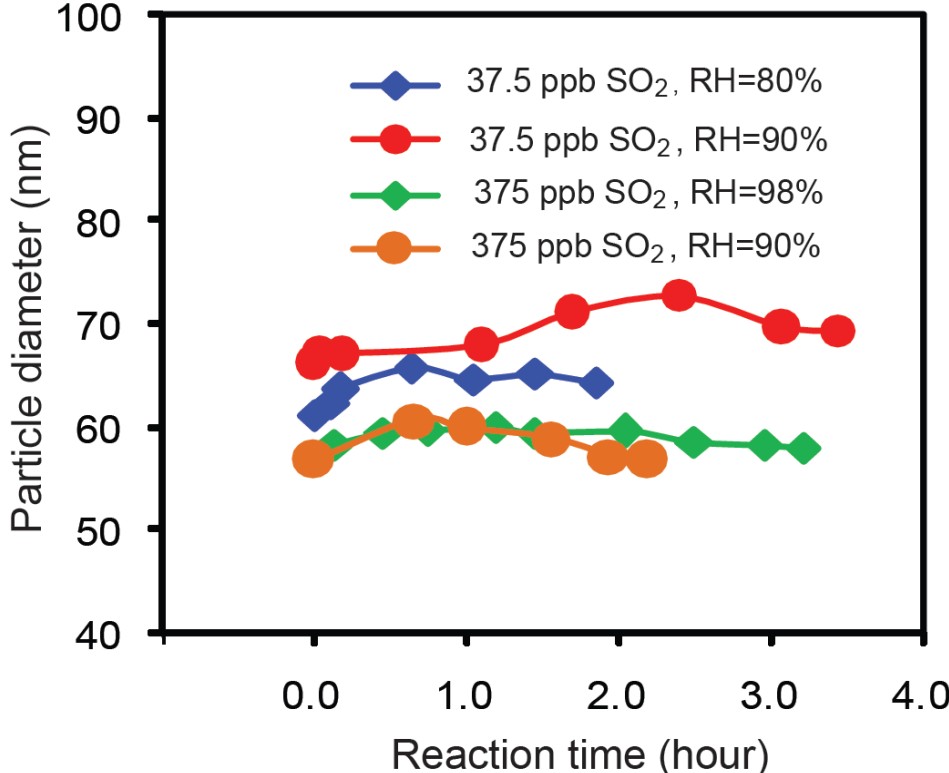

Figure 1. Size evolution of ammonium sulfate particles after exposure to $SO_2$, $NO_2$, and $NH_3$ at
different RH levels. Variations in mobility diameter ($D_p$) of ammonium sulfate particles as a
function of reaction time. The symbols with different colors denote measurements with exposure
to different $SO_2$ concentrations and RH levels. In all cases, the $NO_2$ concentration is 375 ppb,
and the $NH_3$ concentration is 500 ppb.




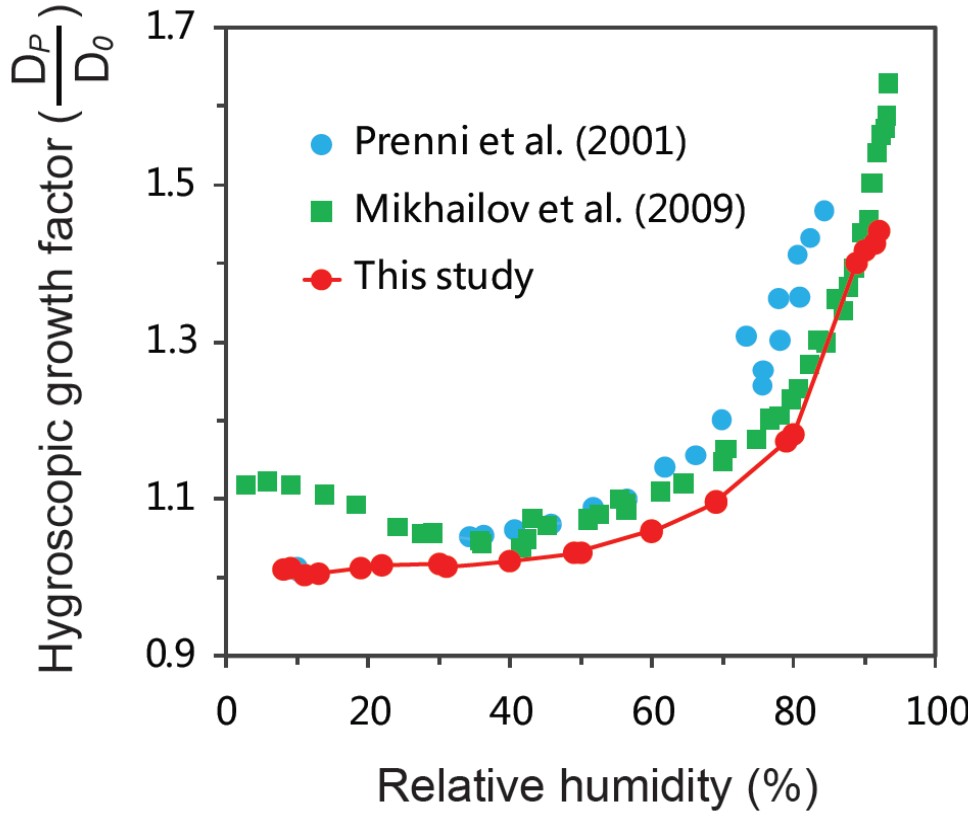

Figure 2. Measured hygroscopic growth factor (HGF) of oxalic acid particles at different RH
conditions. $D_p$ is the particle diameter at an elevated RH, and $D_0$ (100nm) is the initial diameter
of oxalic acid particles at RH = 8%.
.





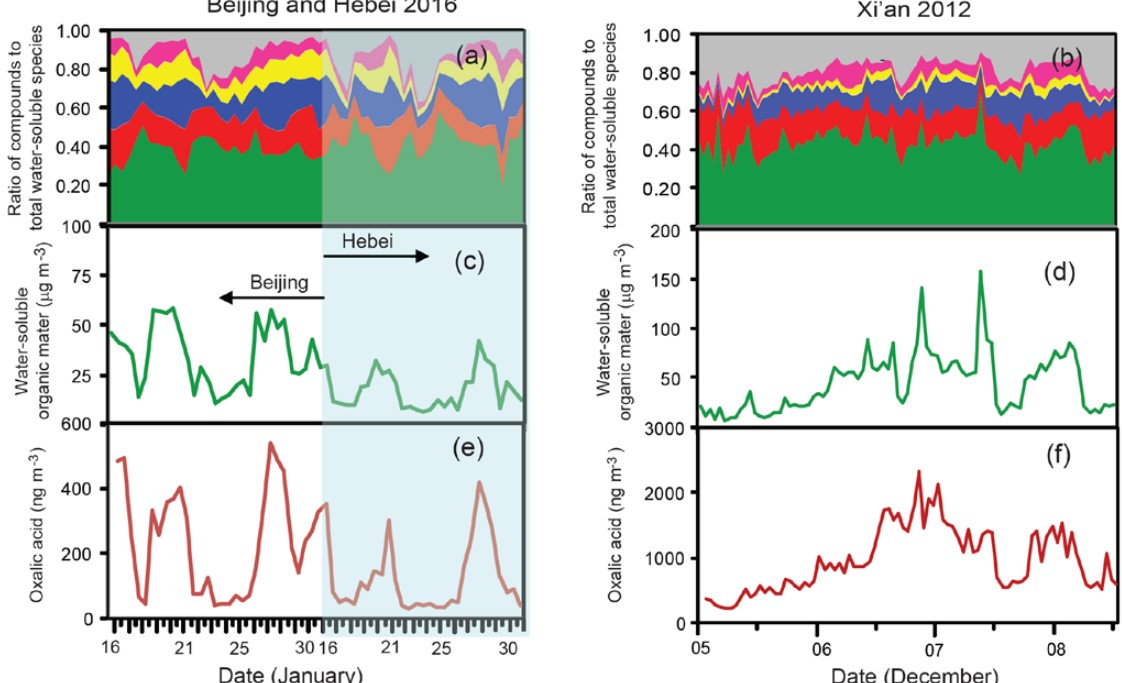

Figure 3. Measurements of water-sluble organic matter (WSOM) of PM$_{2.5}$ collected in Beijing
and Hebei Province during the winter of 2016 (left panels: a, c and e) and in Xi'an during the
winter of 2012 (right panels: b, d and f). In (a) and (b), the green, red, blue, yelow, pink, and
gray colors represent WSOM, sulfate, nitrate, ammonium, chloride, and the others (i.e., the sum
of Na$^+$ + Ca$^{2+}$ + Mg$^{2+}$ + K$^+$), respectively.





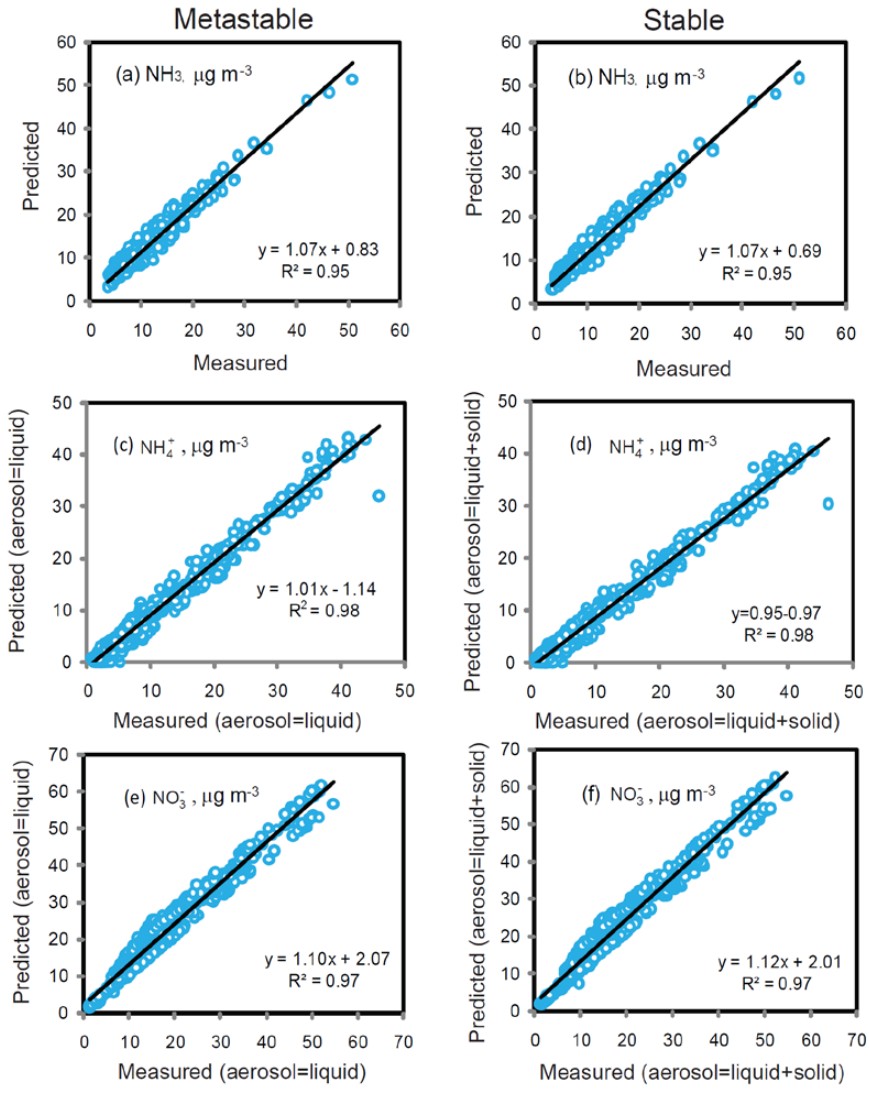


Figure 4. Comparison of measured $NH_3$, $NH_4^+$, and $NO_3^-$ concentrations with those predicted by
ISORROPIA-II model using the forward mode under the metastable (left panels) and stable
assumptions (right panels).