# Peer review of "Particle acidity and sulfate production during severe haze events in China cannot be reliably inferred by assuming a mixture of inorganic salts"

_Atmospheric Chemistry and Physics, 2018_

## Referee Comment (RC1) · Anonymous Referee #1 · 5 Apr 2018

The manuscript presents field and laboratory results of sulfate formation during winter haze conditions in China. It points to the potentially important role that organics play in aerosol acidity and hygroscopic growth in China. An important finding is that SO2 oxidation by NO2, a new mechanism proposed for Beijing winter haze, is significant only on oxalic acid as seed particles, but not on ammonium sulfate due to the strong acidity. This finding was not stressed in the earlier work of the authors (Wang et al. 2016, PNAS). The paper is well written for most parts and should be published in ACP after the following comments are addressed.

My main comment is the representativeness of the lab experiment for atmospheric con-

ditions in winter haze in China. I would expect the organics and inorganic compounds are mixed in ambient aerosol particles, so there will not exist pure oxalic acid aerosol or pure ammonium sulfate aerosol as used in the chamber experiments. I would suggest the authors discuss this issue in the manuscript, particularly with regard to how their proposed mechanism should be adopted in models where aerosols are typically assumed internally mixed.

Specific comments: Figure 1 shows no particle growth with ammonium sulfate as the seed particle. It would be convenient to compare the same growth chart with oxalic acid as the seed particle.

Figure 2: the graph shows growth of oxalic acid seed particles in the chamber experiments at different RH. How much of the growth is due to water and how much is it due to sulfate formation?

Figure 3. Please add sulfate time series and discuss if sulfate correlates with WSOC and/or oxalic acid.

[Figure]

---

## Referee Comment (RC2) · Anonymous Referee #2 · 18 Apr 2018

The manuscript "Particle acidity and sulfate production during severe haze events in China cannot be reliably inferred by assuming a mixture of inorganic salts" of Gehui Wang et al. presents laboratory measurements on the hygroscopicity and particle growth of selected particles under controlled conditions. The work contributes to the current understanding of the aqueous oxidation of SO2 by NO2. The authors also report on field measurements of the chemical composition of the water soluble fraction of fine PM in three sites in China during winter episodes. The results suggest little sulfate production in aqueous ammonium sulfate particles due to high acidity, whereas high sulfate production is observed from oxalic acid particles due to low acidity. The manuscript is well written and the results are relevant and I recommend the publication

of the manuscript after the following aspects are addressed.

In Section 2, add a detailed description of the sampling conditions during the field measurements for the three sites in terms of meteorology (with emphasis on RH and T), sampling size, and a qualitative description of the emission sources potentially impacting the observations. In Section 4, discuss whether any of these sampling conditions can also contribute to the observed differences.

How similar are the chemical and thermodynamic conditions in the laboratory measurements to actual haze conditions during the field measurements? The authors should also explicitly address the caveats of extrapolating the laboratory results in terms of mixing state, acidity, and chemical organic composition (organic mass factions are typically large during these events).

The results in figure 2 show a large increase in the hygroscopic growth factor of oxalic acidy particles when increasing RH. Can the authors discuss possible reasons why these results differ from those of Peng et al., 2001 under high RH?
* * *

---

## Short Comment (SC1) · 19 Apr 2018

Short Comments on "*Particle acidity and sulfate production during severe haze events in China cannot be reliably inferred by assuming a mixture of inorganic salts*" by Wang et al. (2018)

Comments submitted by Shaojie Song, songs@seas.harvard.edu

Wang et al. (2018) discussed particle acidity and sulfate production for haze events in China by laboratory and field experiments. I think this paper may benefit from the following general and specific comments.

10 **A general comment on laboratory experiments**

I find this paper's discussion on the laboratory experiments of sulfate production and particle acidity confusing. In Section 3.1, this paper wrote, "*The ammonium oxalate is less acidic than ammonium sulfate*". An example was given to support this statement: "*the pH value of 0.1 M ammonium oxalate is*
15 *6.5, which is one unit higher than that of ammonium sulfate (the pH value of 0.1 M (NH₄)₂SO₄ solution is 5.5)*". This paper also suggested that the distinct sulfate production rates in Wang et al. (2016) and (2018) are due to the distinct acidity for ammonium oxalate seed and ammonium sulfate seed particles. The abstract of this paper wrote, "*Ammonium sulfate and oxalic acid seed particles exposed to vapors of SO₂, NO₂, and NH₃ at high relative humidity (RH) exhibit distinct size growth and sulfate formation. Aqueous*
20 *ammonium sulfate particles exhibit little sulfate production because of high acidity, in contrast to aqueous oxalic acid particles with significant sulfate production because of low acidity*".

**I think that the statement, "*The ammonium oxalate is less acidic than ammonium sulfate*", is true for the bulk aqueous solution, but false for the aerosol particles under the experimental conditions in**
25 **Wang et al. (2016) and (2018).** The major difference between the bulk aqueous solution and aerosol particles is in their time scale of reaching the thermodynamic equilibrium with the gas phase. The time required for the thermodynamic equilibration of 100 nm aerosol particles should be very short (in minutes or less), while this characteristic time of bulk aqueous solution is much longer. Thus, we do not need consider the issue of thermodynamic equilibrium when determining the acidity of bulk aqueous solution
30 but usually we have to consider this for aerosol particles.

[Figure]

Bulk aqueous solution
10 cm

Aerosol particles
100 nm

Using the E-AIM thermodynamic equilibrium model (*http://www.aim.env.uea.ac.uk/aim/aim.php*), it is easy to calculate that the pH values of 0.1 M ammonium oxalate and 0.1 M ammonium sulfate bulk aqueous solutions are 6.4 and 5.4, respectively, at 298 K. This is obvious, as the acidity of sulfuric acid ($pK_a = -3$, 1.99) is stronger than that of oxalic acid ($pK_a = 1.25$, 4.14). I also calculated the pH values for aerosol particles with the E-AIM model under the following situations (which should be consistent with laboratory experiments by Wang et al.): temperature $= 298$ K, RH $= 90\%$, $NH_3 = 20$ µmol m$^{-3}$ air, $(NH_4)_2SO_4 = 0.1$ µmol m$^{-3}$ air, and the levels of oxalic acid increase from 0 to 0.32 µmol m$^{-3}$ air (representing the mixing of ammonium oxalate and ammonium sulfate). One characteristic of the experimental conditions is that $NH_3$ is super rich, that is to say, $[NH_3] >> [SO_4^{2-}] + [C_2O_4^{2-}]$, and therefore a large amount of $NH_3$ can exist in the gas phase and buffer the pH of the aqueous solution. The equilibrium of dissolution and dissociation of ammonia in water can be expressed as: $NH_{3(g)} + H^+_{(aq)} \leftrightarrow NH_4^+_{(aq)}$. The figure below shows that the calculated particle pH values do not vary with different mixing ratios of ammonium sulfate and ammonium oxalate. Thus, the statement in this paper, "*aqueous ammonium oxalate/(NH4)2SO4 particles exhibit a lower acidity than that of (NH4)2SO4 particles*" does not stand.

[Figure]

Variation of particle pH with different mixing ratios of ammonium sulfate and ammonium oxalate

The above E-AIM model calculations suggest that the pH values for aerosol particles of ammonium oxalate and ammonium sulfate, under the experimental conditions, should be nearly the same (difference < 0.02 pH unit). Then, **the interesting question is: why the sulfate production is different for the oxalic acid and ammonium sulfate seed particles in the laboratory experiments**? I learn from Wang et al. (2016) and (2018) that these experiments have "*comparable concentrations for $SO_2$, $NO_2$, and $NH_3$*" and the aerosol particles are "*in the same phase-state (aqueous)*". It would be great if the authors can provide more insights into this question. The inputs of these E-AIM model calculations are attached at the end of this file, and I am happy to conduct additional thermodynamic model calculations, if requested.

**Specific comments:**

Page 8, Line 180: The solubility of $NO_2$ should not change with particle acidity as it does not dissociate.

Page 11, Lines 244-259: On the pH predictions under the metastable and stable mode, this paper wrote, "*More recently, it was suggested that the large discrepancy in predicting pH is attributable to the differences in the model assumptions (Song et al., 2018)*". The citation seems not clear. This Song et al. (2018) study demonstrated that there were coding errors in the stable mode of the ISORROPIA-II standard model, and that the assumed particle phase states do not significantly impact pH predictions. The pH values of 6.96±1.33 under the stable mode in Xi'an winter 2012 were affected these coding errors.

Page 11, Lines 261-265: "*Guo et al. (2017) and Liu et al. (2017) assumed negligible particle water associated with the organic aerosol mass. Such an assumption is clearly invalid since aerosols typically contain a large portion of WSOM in China, including organic nitrogen species and acids.*" The particle water associated with organics should be much smaller compared to that associated with inorganic salts, given the small hygroscopicity parameter (κ) of organics. A very recent paper, Wu et al. (2018), estimated aerosol water contents under Beijing winter haze conditions, by the ISORROPIA-II model using inorganic salts measurements and by the combination of the measured size-resolved hygroscopic growth factors and particle number size distributions, and showed that these two methods agreed well with each other. This Wu et al. (2018) study demonstrated the minor contribution of organic compounds to aerosol water contents.

80    **Reference**

Song, S., Gao, M., Xu, W., Shao, J., Shi, G., Wang, S., Wang, Y., Sun, Y., and McElroy, M. B.: Fine particle pH for Beijing winter haze as inferred from different thermodynamic equilibrium models, Atmos. Chem. Phys. Discuss., doi:10.5194/acp-2018-6, in review, 2018.

Wang, G., Zhang, R., Gomez, M. E., Yang, L., Levy Zamora, M., Hu, M., Lin, Y., Peng, J., Guo, S.,

85    Meng, J., Li, J., Cheng, C., Hu, T., Ren, Y., Wang, Y., Gao, J., Cao, J., An, Z., Zhou, W., Li, G., Wang, J., Tian, P., Marrero-Ortiz, W., Secrest, J., Du, Z., Zheng, J., Shang, D., Zeng, L., Shao, M., Wang, W., Huang, Y., Wang, Y., Zhu, Y., Li, Y., Hu, J., Pan, B., Cai, L., Cheng, Y., Ji, Y., Zhang, F., Rosenfeld, D., Liss, P. S., Duce, R. A., Kolb, C. E., and Molina, M. J.: Persistent sulfate formation from London Fog to Chinese haze, Proc. Natl. Acad. Sci. U.S.A., 113, 13630-13635, doi:10.1073/pnas.1616540113, 2016.

90    Wang, G., Zhang, F., Peng, J., Duan, L., Ji, Y., Marrero-Ortiz, W., Wang, J., Li, J., Wu, C., Cao, C., Wang, Y., Zheng, J., Secrest, J., Li, Y., Wang, Y., Li, H., Li, N., and Zhang, R.: Particle acidity and sulfate production during severe haze events in China cannot be reliably inferred by assuming a mixture of inorganic salts, Atmos. Chem. Phys. Discuss., 2018, 1-23, doi:10.5194/acp-2018-185, 2018.

Wu, Z., Wang, Y., Tan, T., Zhu, Y., Li, M., Shang, D., Wang, H., Lu, K., Guo, S., Zeng, L., and Zhang,

95    Y.: Aerosol liquid water driven by anthropogenic inorganic salts: implying its key role in haze formation over the North China Plain, Environ. Sci. Technol. Lett., 5, 160-166, doi:10.1021/acs.estlett.8b00021, 2018.

**Table. Summary of inputs for the E-AIM (version II) model calculations. The temperature in all of the problems is 298 K. The concentration units of input chemical species are mol m⁻³ air or μmol m⁻³ air.**

| Problem No. | Type | Input |
|---|---|---|
| 1 | Bulk aqueous solution, 0.1 M ammonium oxalate | $H_2O = 55.6$ mol m$^{-3}$, $NH_4^+ = 0.2$ mol m$^{-3}$, $H_2C_2O_4 = 0.1$ mol m$^{-3}$ |
| 2 | Bulk aqueous solution, 0.1 M ammonium sulfate | $H_2O = 55.6$ mol m$^{-3}$, $NH_4^+ = 0.2$ mol m$^{-3}$, $SO_4^{2-} = 0.1$ mol m$^{-3}$ |
| 3 | Aerosol particles | RH = 90%, $NH_3 = 20$ μmol m$^{-3}$, $SO_4^{2-} = 0.1$ μmol m$^{-3}$, $H_2C_2O_4 = 0$ μmol m$^{-3}$ |
| 4 | Aerosol particles | RH = 90%, $NH_3 = 20$ μmol m$^{-3}$, $SO_4^{2-} = 0.1$ μmol m$^{-3}$, $H_2C_2O_4 = 0.01$ μmol m$^{-3}$ |
| 5 | Aerosol particles | RH = 90%, $NH_3 = 20$ μmol m$^{-3}$, $SO_4^{2-} = 0.1$ μmol m$^{-3}$, $H_2C_2O_4 = 0.02$ μmol m$^{-3}$ |
| 6 | Aerosol particles | RH = 90%, $NH_3 = 20$ μmol m$^{-3}$, $SO_4^{2-} = 0.1$ μmol m$^{-3}$, $H_2C_2O_4 = 0.04$ μmol m$^{-3}$ |
| 7 | Aerosol particles | RH = 90%, $NH_3 = 20$ μmol m$^{-3}$, $SO_4^{2-} = 0.1$ μmol m$^{-3}$, $H_2C_2O_4 = 0.08$ μmol m$^{-3}$ |
| 8 | Aerosol particles | RH = 90%, $NH_3 = 20$ μmol m$^{-3}$, $SO_4^{2-} = 0.1$ μmol m$^{-3}$, $H_2C_2O_4 = 0.16$ μmol m$^{-3}$ |
| 9 | Aerosol particles | RH = 90%, $NH_3 = 20$ μmol m$^{-3}$, $SO_4^{2-} = 0.1$ μmol m$^{-3}$, $H_2C_2O_4 = 0.32$ μmol m$^{-3}$ |

---

## Short Comment (SC2) · 19 Apr 2018

*Comment on Wang et al. (by A.Nenes, R.Weber, H.Guo, A.Russell, and P.Vasilakos)*
We feel that the manuscript contains conceptual issues that need to be addressed. Below is what we consider the main ones.

- The authors claim that thermodynamic models cannot be applied to ambient aerosol, because the system is "too complicated" due to the presence of organics. This discredits decades of research and aerosol model development and the extensive literature to evaluate them, without much proof on behalf of the authors. Instead, the authors chose, without supporting evidence, to come up with a much simpler (and considerably less accurate and evaluated) **conceptual model** of aerosol acidity, one that presumably uses the presence of aerosol oxalate as the "smoking gun" of aerosol neutrality, and the proposed $NO_2$-enhanced sulfate production mechanism of Wang et al. (2016). This conceptual model, a fundamental premise of the paper, is incorrect; one can show (using the approach of Meskhidze et al. 2003 and Guo et al. 2016) that a considerable fraction of oxalic acid can partition to strongly acidic aerosol (see figure below). In a manuscript we have just submitted to ACPD (Nah et al., submitted), we use comprehensive field data to verify that the semivolatile partitioning of oxalic acid follows thermodynamic predictions just like for semivolatile inorganic ions, **and is in agreement with modeled pH**.

[Figure]

**Figure:** Fraction of total Oxalic acid (particle/[particle+gas]) that partitions to the aerosol phase as a function of aerosol pH. Note that there is always some oxalate in the aerosol phase, even at very low pH (where it is undissociated), due to the high solubility (Henry's Law constant) of oxalate. Calculations based on Nah et al. (submitted).

- Noteworthy is that Song et al (2018) carried out pH calculations (with the E-AIM model) for Beijing conditions and found that oxalic acid does not modulate aerosol pH considerably. This is in part because oxalate ion constitutes a relatively small fraction of the total ions in solution. The data presented in Wang et al. also shows that relatively minor amounts of oxalate are in the aerosol, so pH cannot be strongly affected. Regarding the discussion on the other effects of organics on pH predictions (such as liquid-liquid phase separation effects and water uptake), please also refer to Pye et al. (2018) and the relevant discussion of Guo et al. (2015).

- *Partitioning calculations for evaluating pH predictions*: The authors raise issues with the use of comparisons between predicted and measured partitioning of semivolatile species as a way to assess the thermodynamic model performance. Here we attempt to clarify the issue.

1. **At high RH**, it is expected that the aerosol is **fully deliquesced**. If this is the case, then all of the measured aerosol ammonium, nitrate and chloride are dissolved in water and in equilibrium with the gas phase. Errors in $H^+$ therefore directly affect the measured inorganic aerosol species and their partitioning. This is when the comparison of predicted vs. measured partitioning most strongly constrains aerosol pH. The published body of evidence (e.g., Guo et al., 2015; Liu et al., 2017a; Song et al., 2018) shows that modeled pH values *are* reliable and representative of the aerosol acidity.

2. **At low RH**, the situation is less straightforward as the aerosol **may still be deliquesced (metastable mode) or be partially in solid state (stable mode)** so not all of the measured aerosol ammonium, nitrate and chloride may be dissolved. In this case, **liquid water content measurements are necessary** for constraining the amount of liquid phase. Guo et al. (2015) did this when supporting their calculations for the SE US; in the case of China, particle bounce measurements by Liu et al., (2017b) show particles are in the liquid state down to low humidity and Wu et al. (2018) found ISORROPIA metastable-predicted liquid water in good agreement with H-TDMA inferred liquid water during winter haze condition in Beijing. All this supports the metastable state for the haze fine particles.

When aerosol water measurements *are not* available, one can still use the RH history of the air mass to assess whether metastable aerosol is favored based on established knowledge of the efflorescence of the major salts that form in the aerosol; in China, the metastable assumption is expected to hold for a wide range of RH, as you rarely go below the efflorescence RH of pure ammonium sulfate (~35%) after exposure – usually during nighttime - to high RH (above 80%, the deliquescence RH of ammonium sulfate); if one considers the large amounts of nitrate that coexist – which effloresces at a *much lower* RH than ammonium sulfate – then you expect the metastable state to dominate for an even wider range of RH, certainly for the humidity levels experienced during intense haze events (e.g., supplementary material of Wang et al., 2016).

Song et al. (2018) finally show that the difference in pH between stable and metastable states may not significantly differ when the two solutions drastically differ in their predicted liquid water (one still requires however evaluation of the pH against partitioning measurements to believe them!). Therefore, asserting that uncertainty in which phase state is assumed invalidates the pH prediction is without merit.

**References**

Guo, H., et al.: Fine-particle water and pH in the southeastern United States, Atmos. Chem. Phys., 15, 5211-5228, doi: 10.5194/acp-15-5211-2015, 2015.

Guo, H., Sullivan, A. P., Campuzano-Jost, P., Schroder, J. C., Lopez-Hilfiker, F. D., Dibb, J. E., Jimenez, J. L., Thornton, J. A., Brown, S. S., Nenes, A., and Weber, R. J.: Fine particle pH and the partitioning of nitric acid during winter in the northeastern United States, J. Geophys. Res., 121, 10,355-310,376, doi: 10.1002/2016jd025311, 2016.

Liu, M., Song, Y., Zhou, T., Xu, Z., Yan, C., Zheng, M., Wu, Z., Hu, M., Wu, Y., and Zhu, T.: Fine particle pH during severe haze episodes in northern China, Geophys. Res. Lett., 44, 5213-5221, 10.1002/2017gl073210, 2017a.

Liu, Y., Z. Wu, Y. Wang, Y. Xiao, F. Gu, J. Zheng, T. Tan, D. Shang, Y. Wu, L. Zeng, M. Hu, A. P. Bateman, and S. T. Martin, Submicrometer Particles Are in the Liquid State during Heavy Haze Episodes in the Urban Atmosphere of Beijing, China, *Envir. Sci Technol. Lett.*, *DOI: 10.1021/acs.estlett.7b00352*, 2017b.

Nah, T., Guo, H., Sullivan, A. P., Chen, Y., Tanner, D. J., Nenes, A., Russell, A., Ng,, N. L., Huey, L. G. and Weber, R. J., Characterization of Aerosol Composition, Aerosol Acidity and Organic Acid Partitioning at an Agriculture-Intensive Rural Southeastern U.S. Site, submitted to Atmos. Chem. Phys.

Pye, H. O. T., Zuend, A., Fry, J. L., Isaacman-VanWertz, G., Capps, S. L., Appel, K. W., Foroutan, H., Xu, L., Ng, N. L., and Goldstein, A. H.: Coupling of organic and inorganic aerosol systems and the effect on gas–particle partitioning in the southeastern US, Atmos. Chem. Phys., 18, 357-370, doi:10.5194/acp-18-357-2018, 2018.

Song, S., Gao, M., Xu, W., Shao, J., Shi, G., Wang, S., Wang, 937 Y., Sun, Y., and McElroy, M. B.: Fine particle pH for Beijing winter haze as inferred from different thermodynamic equilibrium models, Atmos. Chem. Phys. Discuss., 2018, 1-26, 10.5194/acp-2018-6, 2018

Wang, G., Zhang, R., Zamora, M. L., Gomez, M. E., Yang, L., Hu, M., Lin, Y., Guo, S., Meng, J., Li, J., Cheng, C., Hu, T., Ren, Y., Wang, Y., Gao, J., Cao, J., An, Z., Zhou, W., J. Wang, Marrero-Ortiz, W., Tian, P., Secrest, J., Peng, J., Du, Z., Jing Zheng, Shang, D., Zeng, L., Shao, M., Wang, W., Huang, Y., Wang, Y., Zhu, Y., Li, Y., Hu, J., Pan, B., Cai, L., Cheng, Y., Rosenfeld, D., Liss, P. S., Duce, R. A., Kolb, C. E., and Molina, M. J.: Persistent Sulfate Formation from London Fog to Chinese Haze, Proc.Nat.Acad.Sci., 113(48), 13630-13635, 2016.

Wu, Z., Wang, Y., Tan, T., Zhu, Y., Li, M., Shang, D., Wang, H., Lu, K., Guo, S., Zeng, L., and Zhang, Y.: Aerosol Liquid Water Driven by Anthropogenic Inorganic Salts: Implying Its Key Role in Haze Formation over the North China Plain, Environ. Sci. Technol. Lett., 5, 160-166, doi: 10.1021/acs.estlett.8b00021, 2018.

---

## Short Comment (SC3) · 19 Apr 2018

One point that needs to be emphasized is that "bulk" (or beaker-scale) arguments, essentially do not consider the critical issue of equilibrium with the gas phase and the implications thereof, as pointed out by S. Song in his comment. The number of moles in the gas phase can be comparable or far exceed those in the aerosol phase; therefore volatilization of compounds (such as ammonium, oxalate, nitrate and chloride) to establish equilibrium can completely change the composition of the aerosol vs. the expectations obtained from beaker-scale thermodynamics. Understanding this unique but often neglected aspect of the aerosol system is critically important. Thermody-

namic models based on the equilibrium assumption consider this, bulk beaker-scale conceptual models do not.

---

## Short Comment (SC4) · 19 Apr 2018

Shaojie Song, songs@seas.harvard.edu

Below are some more comments and resources that may be helpful for clarifying the third point that Nenes et al. raised: "Partitioning calculations for evaluating pH predictions".

Nenes et al. used two field studies conducted in winter Beijing, Liu et al. (2017b) and Wu et al. (2018), to support that the metastable state for the haze fine particles is favorable. I think these measurement results may be misinterpreted. Before the

detection of particle rebound fraction, Liu et al. (2017b) used a RH conditioner to dry the ambient particles to 20% RH and then humidified these particles to the ambient RH. Thus, it is likely that the phase state (no matter what it was) of aerosol particles has been changed to a stable state during the measurements. After converting the measured values of rebound fraction to those of hygroscopic growth factor, I found that the relationship between RH and hydroscopic growth factor agreed better with the predicted relationship using the stable state assumption and the ISORROPIA-II model. On the other hand, Wu et al. (2018) seemed to increase the RH to a value above the deliquescence RH of all the inorganic salts and thus the aerosol water content was actually calculated in a metastable state by the H-TDMA approach, which may partly explain why the "ISORROPIA metastable-predicted liquid water in good agreement with H-TDMA inferred liquid water". In short, these measurements did not provide clear evidence of particle phase state (metastable or stable) since the RH history of ambient aerosol particles has been changed during the field measurements.

I think it is unlikely to "use the RH history of the air mass to assess whether metastable aerosol is favored based on established knowledge of the efflorescence of the major salts that form in the aerosol", due to two factors: (1) typically, the formation of winter haze events in North China is associated with the change of wind direction (from strongly northerly winds to weak southerly winds) and the accumulation of water vapor in the atmosphere (ambient RH increases from about 20% to about 80%); (2) the efflorescence RH is unknown due to the complex aerosol chemical composition (the existence of ammonium nitrate may reduce efflorescence RH while the existence of dust materials may increase the efflorescence RH). In short, it is unlikely to figure out particle phase states from theoretical calculations because of the very large variability of ambient RH and the difficulty in estimating the efflorescence RH for multicomponent salt.

Before we have enough evidence to demonstrate what the phase state of aerosol particles is, a practical approach I believe is to predict pH for both stable and metastable

states, which can provide an estimate of its uncertainty due to the phase state assumption.

Some additional related discussions can be found in my paper under review in ACPD:

https://www.atmos-chem-phys-discuss.net/acp-2018-6/

https://www.atmos-chem-phys-discuss.net/acp-2018-6/acp-2018-6-AC2-supplement.pdf

I thank Pengfei Liu (Harvard) and Zhijun Wu (PKU) very much for their help in understanding the measurement principles of rebound fraction and hygroscopicity parameter.

References

Liu, Y., Z. Wu, Y. Wang, Y. Xiao, F. Gu, J. Zheng, T. Tan, D. Shang, Y. Wu, L. Zeng, M. Hu, A. P. Bateman, and S. T. Martin, Submicrometer Particles Are in the Liquid State during Heavy Haze Episodes in the Urban Atmosphere of Beijing, China, Envir. Sci Technol. Lett., DOI: 10.1021/acs.estlett.7b00352, 2017b

Wu, Z., Wang, Y., Tan, T., Zhu, Y., Li, M., Shang, D., Wang, H., Lu, K., Guo, S., Zeng, L., and Zhang, Y.: Aerosol Liquid Water Driven by Anthropogenic Inorganic Salts: Implying Its Key Role in Haze Formation over the North China Plain, Environ. Sci. Technol. Lett., 5, 160-166, doi: 10.1021/acs.estlett.8b00021, 2018.

———————————————————

---

## Author Comment (AC1) · 26 May 2018

General comments:

(1) Comments: The manuscript presents field and laboratory results of sulfate formation during winter haze conditions in China. It points to the potentially important role that organics play in aerosol acidity and hygroscopic growth in China. An important finding is that SO2 oxidation by NO2, a new mechanism proposed for Beijing winter haze, is significant only on oxalic acid as seed particles, but not on ammonium sulfate

due to the strong acidity. This finding was not stressed in the earlier work of the authors (Wang et al. 2016, PNAS). The paper is well written for most parts and should be published in ACP after the following comments are addressed.

Response: We thank the reviewer for the comments above.

(2) Comments: My main comment is the representativeness of the lab experiment for atmospheric conditions in winter haze in China. I would expect the organics and inorganic compounds are mixed in ambient aerosol particles, so there will not exist pure oxalic acid aerosol or pure ammonium sulfate aerosol as used in the chamber experiments. I would suggest the authors discuss this issue in the manuscript, particularly with regard to how their proposed mechanism should be adopted in models where aerosols are typically assumed internally mixed.

Response: We agree with the reviewer that organic and inorganic compounds are mixed in ambient aerosol particles, and there are no pure oxalic acid aerosols or pure ammonium sulfate aerosols in the atmosphere. In fact, in our chamber experiment the seeded particles were initially pure oxalic acid and became a mixture of ammonium oxalate and ammonium sulfate with the proceeding of the formation of sulfate on the seeded oxalic acid particles. Based on our laboratory simulation, we calculated the uptake coefficient $SO_2$ on oxalic acid particles in the reaction chamber, which is $8.3\pm5.7\times10^{-5}$ (Table S6, Wang et al., 2016) under the humid conditions and consistent with that $(4.5\pm1.1\times10^{-5})$ (Table S6, Wang et al., 2016) observed in Beijing during the haze period of 2015. Such a consistence indicates that our chamber results including the uptake coefficient would be applicable for model simulation. We added those related discussions into the text. See page 9, line 199-202 and page 13-14, line 313-322.

Specific comments:

(3) Comments: Figure 1 shows no particle growth with ammonium sulfate as the seed particle. It would be convenient to compare the same growth chart with oxalic acid as the seed particle.

Response: Suggestion taken. We added the growth of the seeded oxalic acid particles into the figure for the comparison. See Figure 2 and related statements in page 7, line 167-169.

(4) Comments: Figure 2: the graph shows growth of oxalic acid seed particles in the chamber experiments at different RH. How much of the growth is due to water and how much is it due to sulfate formation?

Response: Figure 2 shows the hygroscopic growth factor of oxalic acid measured by a HTDMA system, which is not the growth of oxalic acid seed particles in the chamber experiment.

(5) Comments: Figure 3. Please add sulfate time series and discuss if sulfate correlates with WSOC and/or oxalic acid.

Response: Suggestion taken. We have revised the figure by adding sulfate time series (see Figure 3 g and h). As shown in Figure 3g and h, during the field observation periods sulfate at the three sites showed a temporal variation pattern similar to that of oxalic acid with a robust linear correlation ($r^2$=0.67, 0.84 and 0.61 in Xi'an, Beijing and Hebei Province, respectively). Such a correlation was also reported by other researchers (Wang et al, 2017, Yu et al, 2005), suggesting the cooccurrence of both compounds in atmospheric aerosols. The related discussion was added into the revised manuscript. See page 10, line 227-235.

References

Wang, G., Zhang, R., Zamora, M. L., Gomez, M. E., Yang, L., Hu, M., Lin, Y., Guo, S., Meng, J., Li, J., Cheng, C., Hu, T., Ren, Y., Wang, Y., Gao, J., Cao, J., An, Z., Zhou, W., Jiayuan Wang, Marrero-Ortiz, W., Tian, P., Secrest, J., Peng, J., Du, Z., Jing Zheng, Shang, D., Zeng, L., Shao, M., Wang, W., Huang, Y., Wang, Y., Zhu, Y., Li, Y., Hu, J., Pan, B., Cai, L., Cheng, Y., Rosenfeld, D., Liss, P. S., Duce, R. A., Kolb, C. E., and Molina, M. J.: Persistent Sulfate Formation from London Fog to Chinese Haze,

Proceedings of National Academy of Science of United States of America, 113(48), 13630-13635, doi/13610.11073/pnas.1616540113, 2016.

Wang, J., Wang, G., Gao, J., Wang, H., Ren, Y., Li, J., Zhou, B., Wu, C., Zhang, L., Wang, S., and Chai, F.: Concentrations and stable carbon isotope compositions of oxalic acid and related SOA in Beijing before, during, and after the 2014 APEC, Atmospheric Chemistry and Physics, 17(2), 981-992, 2017.

Yu, J. Z., Huang, X. F., Xu, J. H., and Hu, M.: When aerosol sulfate goes up, so does oxalate: Implication for the formation mechanisms of oxalate, Environmental Science & Technology, 39(1), 128-133, 2005.

---

## Author Comment (AC2) · 26 May 2018

General comments:

(1) Comments The manuscript "Particle acidity and sulfate production during severe haze events in China cannot be reliably inferred by assuming a mixture of inorganic salts" of Gehui Wang et al. presents laboratory measurements on the hygroscopicity and particle growth of selected particles under controlled conditions. The work contributes to the current understanding of the aqueous oxidation of SO2 by NO2. The

authors also report on field measurements of the chemical composition of the water soluble fraction of fine PM in three sites in China during winter episodes. The results suggest little sulfate production in aqueous ammonium sulfate particles due to high acidity, whereas high sulfate production is observed from oxalic acid particles due to low acidity. The manuscript is well written and the results are relevant and I recommend the publication of the manuscript after the following aspects are addressed.

Response: We thank the reviewer's comments above.

Specific comments

(1) Comments: In Section 2, add a detailed description of the sampling conditions during the field measurements for the three sites in terms of meteorology (with emphasis on RH and T), sampling size, and a qualitative description of the emission sources potentially impacting the observations. In Section 4, discuss whether any of these sampling conditions can also contribute to the observed differences.

Response: Suggestion taken. We have added the detailed descriptions on the sampling conditions at the three sites including RH, temperature, particle cutoff size and a qualitative description of the potential emissions sources. See the above information on page 6, line 135-140. We also discussed whether these sampling conditions can also contribute to the observed differences. See page 9, line 219-221.

(2) Comments: How similar are the chemical and thermodynamic conditions in the laboratory measurements to actual haze conditions during the field measurements? The authors should also explicitly address the caveats of extrapolating the laboratory results in terms of mixing state, acidity, and chemical organic composition (organic mass factions are typically large during these events).

Response: The concentrations of gases we used for the chamber simulation are about 10 times higher than those in Beijing, but their relative abundances and the relative humidity are similar to those during the field measurements. As discussed above,

the uptake coefficient of SO2 on oxalic acid particles extrapolated from our laboratory measurements is similar to that in Beijing during the observation period, which suggests that our proposed mechanism is applicable to the actual haze conditions in China. Based on the field measurement and model simulation, Cheng et al (2016) also reported a similar result, corroborating our lab work results. As reply to referee #1, we have added related discussions into the text, see page 9, line 199-202, and page 14, line 316-322.

(3) Comments: The results in figure 2 show a large increase in the hygroscopic growth factor of oxalic acidy particles when increasing RH. Can the authors discuss possible reasons why these results differ from those of Peng et al., 2001 under high RH?

Response: The difference of hygroscopic growth factors of oxalic acid particles between Peng et al 2001 and this study is mainly due to the different methods used for the hygroscopicity measurement. In the work of Peng et al (2001), they used an electrodynamic balance (EDB) system to trap and levitate a charged particle. The relative mass of a particle equilibrated at different relative humidities was determined by measuring the balancing voltage. The size of each particle studied was not measured but was estimated from visual inspection using a microscope (Peng et al., 2001). In contrast, the hygroscopicity of oxalic acid in this study was measured by using a hygroscopicity tandem differential mobility analyzer (HTDMA) system. The particle size was measured by a SMPS (scanning mobility particle sizer). The EDB system measured the size of a charged particle through determining its optical property, while the HTDMA system measured by the SMPS system through determining its mobility in the electric field. In addition, the size range of oxalic acid particles measured by Peng et al (2001) is different from that measured by this study. The size range of charged oxalic acid droplet measured by the EDB system is 10-20 microns, while that measured by the HTDMA system in this study is less than 1.0 micron. As shown in Figure 2, our result is very close to that measured by Mikhailov et al (2009) and Prenni et al (2001), who also used a HTDMA system for the hygroscopcity measurement, further suggesting that the

difference between the measured growth factors is due to the different methods. We briefly explained this difference in the revised version, see page 7-8, line 175-180.

References

Cheng, Y., Zheng, G., Wei, C., Mu, Q., Bo Zheng, Wang, Z., Gao, M., Zhang, Q., He, K., Carmichae, G., Pöschl, U., and Su, H.: Reactive nitrogen chemistry in aerosol water as a source of sulfate during haze events in China, Science Advances, 2, e1601530, 2016.

Mikhailov, E., Vlasenko, S., Martin, S. T., Koop, T., and Poeschl, U.: Amorphous and crystalline aerosol particles interacting with water vapor: conceptual framework and experimental evidence for restructuring, phase transitions and kinetic limitations, Atmospheric Chemistry and Physics, 9, 9491-9522, 2009.

Peng, C., Chan, M. N., and Chan, C. K.: The hygroscopic properties of dicarboxylic and multifunctional acids: Measurements and UNIFAC predictions, Environmental Science & Technology, 35(22), 4495-4501, 2001.

Prenni, A. J., DeMott, P. J., Kreidenweis, S. M., Sherman, D. E., Russell, L. M., and Ming, Y.: The effects of low molecular weight dicarboxylic acids on cloud formation, Journal of Physical Chemistry A, 105, 11240-11248, 2001.

Wang, G., Zhang, R., Zamora, M. L., Gomez, M. E., Yang, L., Hu, M., Lin, Y., Guo, S., Meng, J., Li, J., Cheng, C., Hu, T., Ren, Y., Wang, Y., Gao, J., Cao, J., An, Z., Zhou, W., Jiayuan Wang, Marrero-Ortiz, W., Tian, P., Secrest, J., Peng, J., Du, Z., Jing Zheng, Shang, D., Zeng, L., Shao, M., Wang, W., Huang, Y., Wang, Y., Zhu, Y., Li, Y., Hu, J., Pan, B., Cai, L., Cheng, Y., Rosenfeld, D., Liss, P. S., Duce, R. A., Kolb, C. E., and Molina, M. J.: Persistent Sulfate Formation from London Fog to Chinese Haze, Proceedings of National Academy of Science of United States of America, 113(48), 13630-13635, doi/13610.11073/pnas.1616540113, 2016.

[Figure]

2018.

---

## Author Comment (AC3) · 28 May 2018

Song calculated the acidity of the aerosol mixture of ammonium sulfate and ammonium oxalate by using the thermodynamic model and stated that pH of the 0.1 M ammonium sulfate aerosols mixed with different molar ratio of ammonium oxalate is nearly the same (around pH= 4.3, see the figure given by Song). We think this model calculation needs experimental validations. We agree with Shaojie Song and A. Nenes et al that the equilibrium with the gas phase could be significantly different between beaker-scale and aerosols. Due to the tiny size and complex compositions, pH of atmospheric aerosols cannot be measured directly and is, instead, often estimated by

thermodynamic models based on some simplified assumptions. The uncertainty of the pH estimation is significant in some cases. Our lab chamber simulation results showed that SO2 could be efficiently oxidized by NO2 into SO42- on the ammonium oxalate aerosols, which is derived from the reaction of seeded oxalic acid particles with NH3(g), under the high humid conditions with ammonia neutralization, but such a conversion does not efficiently happen on the ammonium sulfate seeds under the same conditions. Such a difference indicates that pH of aqueous ammonium oxalate aerosols is high enough for the SO2 conversion.

We agree with Shaojie Song that the equilibrium of dissolution and dissociation of ammonia in water can be expressed as: NH3(g) + H+(aq) ↔ NH4+(aq), but the concentration of H+ in aqueous aerosols is also equilibrated with all anions, e.g., sulfate (SO42-) and oxalate (C2O42-). Since C2O42- is a weak acid, it would exist in part as C2O4H+/C2O4H2 by combining some H+. Therefore, pH of ammonium oxalate aerosols should be different from that of ammonium sulfate aerosols, which is the reason why sulfate is efficiently produced on oxalic acid particles but not ammonium sulfate particles under high RH conditions with NH3 neutralization.

Our reply to the specific comments

(1) As for the statement related to SO2 and NO2 solubility, we removed the phrase of "NO2 solubility" and revised the sentence.

(2) Song et al claimed that the assumed particle phase states do not significantly impact pH predictions and the higher pH values of 6.9±1.33 under stable mode in Xi'an were affected by the ISORROPIA-II model code errors. As commented by the model developers, A. Nenes et al, this explanation still requires evaluation.

(3) As for the contribution of organics to aerosol water contents, our lab measurement shows that the growth factor of oxalic acid is about 1.5 at a RH90% conditions (Figure 2) (Wang et al., 2018), which is comparable to ammonium sulfate, suggesting that the contribution of organic compounds to aerosol-associated water could be significant in

some cases, and cannot be neglected.

References:

Wang, G., Zhang, F., Peng, J., Duan, L., Ji, Y., arrero-Ortiz, W., Wang, J., Li, J., Wu, C., Cao, C., Wang, Y., Zheng, J., Secrest, J., Li, Y., Wang, Y., Li, H., Li, N., and Zhang, R.: Particle acidity and sulfate production during severe 1 haze events in China cannot be reliably inferred by assuming a mixture of inorganic salts Atmos. Chem. Phys. Discuss., https://doi.org/10.5194/acp-2018-1852018, 2018.

---

## Author Comment (AC4) · 28 May 2018

We thank S. Song for his additional comments on the third point "partitioning for evaluating pH predictions" raised by A. Nenes et al.

---

## Author Comment (AC5) · 28 May 2018

*Comment on Wang et al. (by A.Nenes, R.Weber, H.Guo, A.Russell, and P.Vasilakos)*
**Comments from A. Nenes et al:**
We feel that the manuscript contains conceptual issues that need to be addressed. Below is what we consider the main ones.

The authors claim that thermodynamic models cannot be applied to ambient aerosol, because the system is "too complicated" due to the presence of organics. This discredits decades of research and aerosol model development and the extensive literature to evaluate them, without much proof on behalf of the authors. Instead, the authors chose, without supporting evidence, to come up with a much simpler (and considerably less accurate and evaluated) conceptual model of aerosol acidity, one that presumably uses the presence of aerosol oxalate as the "smoking gun" of aerosol neutrality, and the proposed NO2-enhanced sulfate production mechanism of Wang et al. (2016). This conceptual model, a fundamental premise of the paper, is incorrect; one can show (using the approach of Meskhidze et al. 2003 and Guo et al. 2016) that a considerable fraction of oxalic acid can partition to strongly acidic aerosol (see figure below). In a manuscript we have just submitted to ACPD (Nah et al., submitted), we use comprehensive field data to verify that the semivolatile partitioning of oxalic acid follows thermodynamic predictions just like for semivolatile inorganic ions, and is in agreement with modeled pH.

[Figure]

Figure 1: Fraction of total oxalic acid (particle/[particle+gas]) that partitions to the aerosol phase as a function of aerosol pH. Note that there is always some oxalate in the aerosol phase, even at very low pH (where it is undissociated), due to the high solubility (Henry's Law constant) of oxalate. Calculations based on Nah et al. (submitted).

**Reply** : We thank Dr. Nenes et al for their interesting comments. Before giving the detailed comments, we would like to say that we didn't intend to discredit anyone, instead, we only showed our experimental results and discussed the potential mechanism leading to the rapid formation of sulfate during the haze events in China by simulating heterogeneous oxidation of $SO_2$ by $NO_2$ on oxalic acid seed particles in a smog chamber. Our chamber results can successfully explain the mechanism of the rapid formation of sulfate in China during the wintertime haze periods. The $SO_2$ uptake coefficient calculated from our laboratory chamber

simulations is consistent with that observed in Beijing haze periods (Wang et al., 2016, 2018) but cannot be explained by using the ISORROPIA-II model.

**Comment from A. Nenes et al**

Noteworthy is that Song et al (2018) carried out pH calculations (with the E-AIM model) for Beijing conditions and found that oxalic acid does not modulate aerosol pH considerably. This is in part because oxalate ion constitutes a relatively small fraction of the total ions in solution. The data presented in Wang et al. also shows that relatively minor amounts of oxalate are in the aerosol, so pH cannot be strongly affected. Regarding the discussion on the other effects of organics on pH predictions (such as liquid-liquid phase separation effects and water

**Reply**: The oxalic acid concentration in aerosols calculated by Song (2018) is not minor, which varies from 0.01 $\mu$mol m$^{-3}$ to 0.32 $\mu$mol m$^{-3}$ and is comparable and even higher than sulfate (0.1$\mu$mol m$^{-3}$). Oxalic acid is often found to be the most abundant dicarboxylic acid in the atmosphere. However, water-soluble organic acids in urban atmospheric aerosols consist of hundreds of species with molecular weight ranging from the smallest formic acid to the very big humic acid like compounds. All of these organic acids are responsible for the aerosol acidity. Therefore, one cannot say that pH cannot be strongly affected by organic acid because oxalic acid in the aerosols is minor. As we discussed in the manuscript, acidity of an ambient aerosols is not only determined by inorganic species but also by organic compounds (e.g., organic acid, carbonyls and amines).

**Comment from A. Nenes et al**

*Partitioning calculations for evaluating pH predictions*: The authors raise issues with the use of comparisons between predicted and measured partitioning of semivolatile species as a way to assess the thermodynamic model performance. Here we attempt to clarify the issue……

**Reply**: As for this point, see the short comments from Song (2018)

**Comment from Nenes et al :**

Song et al. (2018) finally show that the difference in pH between stable and metastable states may not significantly differ when the two solutions drastically differ in their predicted liquid water (one still requires however evaluation of the pH against partitioning measurements to believe them!). Therefore, asserting that uncertainty in which phase state is assumed invalidates the pH prediction is without merit.

**Reply**: Song et al (2018) pointed out that pH predicted by ISORROPIA-II model between stable and metastable modes are similar when the model code errors are fixed, and high pH values predicted by the thermodynamic mode under stable mode is due to the model code errors. We think this issue needs a confirmation from the model developer, i.e., Prof. A. Nenes et al. Until now, Guo et al (2017a, 2017b) and many other researchers (e.g., Liu et al., 2017) have claimed that the pH predictions with ISORROPIA-II model by using the metastable mode are remarkably better than that by using the stable mode, on the basis of model evaluation from measured and predicted $NO_3^-$ and $NH_4^+$. However, we found that the predicted concentrations of $NO_3^-$ and $NH_4^+$ by using both the metastable and stable modes are nearly identical (see Wang et al., 2018) but the pH values predicted with these two phase

states are significantly different, which is recently ascribed by Song et al (2018) to the model code errors.

**Comment from Nenes et al** :

One point that needs to be emphasized is that "bulk" (or beaker-scale) arguments, essentially do not consider the critical issue of equilibrium with the gas phase and the implications thereof, as pointed out by S. Song in his comment. The number of moles in the gas phase can be comparable or far exceed those in the aerosol phase; therefore volatilization of compounds (such as ammonium, oxalate, nitrate and chloride) to establish equilibrium can completely change the composition of the aerosol vs. the expectations obtained from beaker-scale thermodynamics. Understanding this unique but often neglected aspect of the aerosol system is critically important. Thermodynamic models based on the equilibrium assumption consider this, bulk beaker-scale conceptual models do not.

**Reply**: We agree on the comment that the equilibrium of a volatile species between the gas and aerosol phases is different from that on a bulk beak-scale. The problem here is that several recent studies using the thermodynamic model estimated the particle acidity and sulfate production rate by treating the PM exclusively as a mixture of inorganic salts dominated by ammonium sulfate and neglected the effects of organic compounds. Noticeably, the estimated pH and sulfate formation rate during pollution periods in China were highly conflicting among the previous studies.

**References**

Guo, H., Weber, R. J., and Nenes, A.: High levels of ammonia do not raise fine particle pH sufficiently to yield nitrogen oxide-dominated sulfate production, Scientific Reports, 7, 12109, doi:12110.11038/s41598-12017-11704-12100, 2017a.

Guo, H., Liu, J., Froyd, K. D., Roberts, J. M., Veres, P. R., Hayes, P. L., Jimenez, J. L., Nenes, A., and Weber, R. J.: Fine particle pH and gas–particle phase partitioning of inorganic species in Pasadena, California, during the 2010 CalNex campaign, Atmospheric Chemistry and Physics, 17, 5703-5719, 2017b.

Liu, M., Song, Y., Zhou, T., Xu, Z., Yan, C., Zheng, M., Wu, Z., Hu, M., Wu, Y., and Zhu, T.: Fine particle pH during severe haze episodes in Northern China, Geophysical Research Letters, 44, 5213-5221 ,doi: 5210.1002/2017GL073210, 2017.

Song, S., Interactive comment on "Particle acidity and sulfate production during severe haze events in China cannot be reliably inferred by assuming a mixture of inorganic salts" by Gehui Wang et al.  Atmos. Chem. Phys. Discuss., https://doi.org/10.5194/acp-2018-185-SC1, 2018.

Song, S., Gao, M., Xu, W., Shao, J., Shi, G., Wang, S., Wang, Y., Sun, Y., and McElroy, M. B.: Fine particle pH for Beijing winter haze as inferred from different thermodynamic equilibrium models, Atmospheric Chemistry and Physics Discussions, https://doi.org/10.5194/acp-2018-6, 2018.

Wang, G., Zhang, R., Zamora, M. L., Gomez, M. E., Yang, L., Hu, M., Lin, Y., Guo, S., Meng, J., Li, J., Cheng, C., Hu, T., Ren, Y., Wang, Y., Gao, J., Cao, J., An, Z., Zhou, W.,

Wang, J., Marrero-Ortiz, W., Tian, P., Secrest, J., Peng, J., Du, Z., Zheng, J., Shang, D., Zeng, L., Shao, M., Wang, W., Huang, Y., Wang, Y., Zhu, Y., Li, Y., Hu, J., Pan, B., Cai, L., Cheng, Y., Rosenfeld, D., Liss, P. S., Duce, R. A., Kolb, C. E., and Molina, M. J.: Persistent Sulfate Formation from London Fog to Chinese Haze, Proceedings of National Academy of Science of United States of America, 113(48), 13630-13635, doi/13610.11073/pnas.1616540113, 2016.

Wang, G., Zhang, F., Peng, J., Duan, L., Ji, Y., arrero-Ortiz, W., Wang, J., Li, J., Wu, C., Cao, C., Wang, Y., Zheng, J., Secrest, J., Li, Y., Wang, Y., Li, H., Li, N., and Zhang, R.: Particle acidity and sulfate production during severe 1 haze events in China cannot be reliably inferred by assuming a mixture of inorganic salts Atmos. Chem. Phys. Discuss., https://doi.org/10.5194/acp-2018-1852018, 2018.